

# δ$^{13}$C carbon isotopic composition of CO₂ in the atmosphere by Lidar. A preliminary study with a CDIAL system at 2-μm.

Fabien Gibert[1], Dimitri Edouart[1], Didier Mondelain[2], Claire Cénac[1], Camille Yver[3]

[1] Laboratoire de Météorologie Dynamique (LMD/IPSL), École polytechnique, Institut polytechnique de Paris, Sorbonne Université, École normale supérieure, PSL Research University, CNRS, École des Ponts, Palaiseau, France
[2] CNRS, LIPhy, Université Grenoble Alpes, Grenoble, France
[3] Laboratoire des Sciences du Climat et de l'Environnement (LSCE/IPSL), Unité mixte CEA-CNRS-UVSQ, UMR8212, 91191 Gif-sur-Yvette, France

*Correspondence to*: Fabien Gibert (gibert@lmd.polytechnique.fr)

**Abstract.** Our understanding of the global carbon cycle needs for new observations of CO₂ concentration at different space and time scales but also would benefit from observations of additional tracers of intra-atmospheric or surface-atmosphere exchanges to characterize sources and sinks. Lidar is a well-known promising technology for this research as it can provide, at the same time, structure of the atmosphere, dynamics and composition of several trace gas concentration. In this framework, a coherent differential absorption lidar (CDIAL) has been developed at LMD to measure simultaneously and separately $^{12}$CO₂ and $^{13}$CO₂ isotopic composition of CO₂ in the atmosphere. It also provides the wind speed along the line of sight of the laser with an additional Doppler ability. This paper investigates the methodology of three wavelengths DIAL in the spectral domain of 2-μm to obtain range-resolved CO₂ isotopic ratio δ$^{13}$C. The set-up of the lidar as well as the signal processing is described in details. First atmospheric measurements along three days are achieved in the surface layer above the suburban area of Ecole Polytechnique campus, Palaiseau, France. Typical performances of the instrument (median values along 70h of measurement) with 10 min of time averaging show: (1) a precision around 0.6% for 1.2 km range resolution for $^{12}$CO₂ mixing ratio (2) a precision around 3.2% for 1.6 km range resolution for $^{13}$CO₂ mixing ratio. In situ co-located gas analyser measurements are used to correct for biases that are explained neither by the spectroscopic database accuracy nor the signal processing and will need further investigation. Nevertheless, this preliminary study enables to make a useful state of the art for current lidar ability to provide δ$^{13}$C measurements in the atmosphere with respect to geophysical expected anomalies and to predict the necessary performances of a future optimized instrument.

## 1 Introduction

CO₂ is the main anthropogenic greenhouse gas responsible for the current global warming. In 2024, its concentration is larger than 410 ppm, a level never reached during the last 2 million years with devasting consequences for present and future life on planet Earth. In our understanding of the carbon cycle, it is fundamental to associate a number of CO₂ molecules with their original surface or atmospheric sources and sinks. In particular, the biospheric sink remains very complex to assess at regional scale with current tools (bottom-up or top-down methods) given the strong ecosystem space and time heterogeneity (Friedlingstein et al., 2020). The mitigation of anthropogenic emissions needs as well other clues than a standard CO₂ mixing ratio measurement to assign a number of molecules to the type of emission: coal, gas, fuel, natural vs anthropogenic.

In this context, the CO₂ stable isotopic fraction δ$^{13}$C is an interesting tracer for CO₂ surface-atmosphere exchanges at local scale. It enables to characterize plant/ soil physiological processes (photosynthesis, respiration, decomposition of organic matter) and may help to issue a diagnosis on sources (anthropogenic, geological) and sinks (biosphere, ocean) of CO₂. In addition, δ$^{13}$C can discriminate the type of plant (C-3 or C-4) and ecosystems that contribute the most to biosphere CO₂ uptake and their evolution with warming conditions (Buchmann et al. 1998; Flanagan et al. 1996). While the difference between C3- and C4-dominated biomes provides the largest observed variation in δ$^{13}$C (Still 2000), even within pure C-3 ecosystems, a



large temporal and spatial variability of $\delta^{13}C$ is observed during nighttime respiration process due to species-specific effects and environmental conditions such as light, temperature and water availability, the latter being a major driver (Buchmann et al. 1997; Brugnoli et al. 1998; Bowling et al. 2002; Ekblad and Hogberg 2001; Pataki et al. 2003; Mortazavi et al. 2005). The natural variation of $\delta^{13}C$ source spreads over a scale of 100 ‰: ~ +5‰ for carbonate-gas $CO_2$ equilibrium in air-sea /geological

water exchanges, -8‰ for standard $CO_2$ in the atmosphere, ~ -14‰ for C-4 plant – air exchanges but ~-27‰ for C-3 plant that is similar to fossil fuel emission (coal and oil) whereas the lowest $\delta^{13}C$ value, ~ -40‰, can be found in gas emission (AIEA, 2008).

Concerning current instrumentation and measurement, tunable diode laser absorption (Cassidy et al. 1982) or recent cavity ring down spectroscopy (Lin et al., 2020) offer interesting ways to make in situ measurements in the atmosphere compared to

former complex mass spectrometry systems (Li et al., 2018). To increase the spatial scale representativity of in situ measurement, an integrated path differential absorption (IPDA) lidar concept at 4.4 µm has also been studied (Shi et al., 2022). Although such system seems to reach similar precision on $\delta^{13}C$ (< 0.2 ‰) than in situ sensor, the horizontal profiling of $\delta^{13}C$ by Lidar will bring outstanding information on sources/sinks pattern and origin. Even the vertical profiling will help to characterize the local/ long distance transport of $CO_2$ in a similar way as for stable water vapor isotopologue Lidar

measurements (Hamperl et al. 2022).

In this framework, a new three wavelengths coherent differential absorption Lidar (CDIAL) at 2 µm has been considered and recently developed at LMD to measure simultaneously $^{12}CO_2$ and $^{13}CO_2$ absorptions. The purpose of this paper is to assess the current performances of the CDIAL system. The first sections describe the methodology and the spectral domain that has been chosen for this study. Then, a following section presents the experimental set-up, especially the new hybrid fibered/bulk laser

source that was developed for this application. The signal processing is described in details to get the first lidar measurements of $^{12}CO_2$ and $^{13}CO_2$ mixing ratio. Current statistical and systematic errors are estimated. A discussion follows where the current performances of $\delta^{13}C$ are confronted to useful geophysical signals to be measured in the atmosphere. Some guidelines for $\delta^{13}C$ efficient lidar measurements are pointed out.

## 2 Methodology

### 2.1 Multiwavelength DIAL theory

To measure the carbon isotopic ratio with the DIAL technique, at least three wavelengths have to be used, one serving as a reference, ideally located in a free-absorption spectral window. The differential absorption coefficients, $\alpha_{i,exp}$, measured at the different wavelengths, are estimated with the mean range gate lidar backscattered signal power, $P_i$, as follows:

$$\alpha_{i,exp} = \frac{d\tau_{i,exp}}{dR} = \frac{d}{dR}\left[-\frac{1}{2} ln\left(\frac{P_i}{P_0}\right)\right] \qquad (1)$$

where $i = 1, 2$ stands for $^{12}CO_2$ and $^{13}CO_2$ wavelengths respectively and $i = 0$ for the reference non-absorbed laser wavelength. $\tau_{i,exp}$ is the measured single path differential absorption optical depth (DAOD). Note that time and space averaging act at different scales in Eq. (1) that will be described later in the dedicated section on signal processing with biases and statistical error analysis. At this point, we just assume: (i) the laser wavelengths are closed enough that aerosol extinction and backscatter variations with wavelength are negligible and (2) the absorption variation is negligible in the range gate used

to measure $P_i$.

The differential absorption coefficients are linked to the trace gas mixing ratios with:

$$\begin{pmatrix} \alpha_{1,exp} \\ \alpha_{2,exp} \end{pmatrix} = \frac{n_{air}}{(1+C_{H2O})}\begin{pmatrix} \Delta\tilde{\sigma}_{12C,1} & \Delta\tilde{\sigma}_{13C,1} & \Delta\tilde{\sigma}_{H2O,1} \\ \Delta\tilde{\sigma}_{12C,2} & \Delta\tilde{\sigma}_{13C,2} & \Delta\tilde{\sigma}_{H2O,2} \end{pmatrix}\begin{pmatrix} C_{12} \\ C_{13} \\ C_{H2O} \end{pmatrix} \qquad (2)$$

where $\Delta\tilde{\sigma}_{12C,i} = \tilde{\sigma}_{12C,i} - \tilde{\sigma}_{12C,0}$ and $\Delta\tilde{\sigma}_{13C,i} = \tilde{\sigma}_{13C,i} - \tilde{\sigma}_{13C,0}$ are the differential absorption cross-section (ACS), $n_{air} = p/(k_B T)$ is the air density with $T$ and $p$ air temperature and pressure and $k_B$ the Boltzman constant, $C_{H2O}$ is the water vapour mixing ratio.





$$\begin{pmatrix} C_{12} \\ C_{13} \end{pmatrix} = \frac{1+C_{H2O}}{Det.n_{air}} \begin{pmatrix} \Delta\tilde{\sigma}_{13C,2} & -\Delta\tilde{\sigma}_{13C,1} \\ -\Delta\tilde{\sigma}_{12C,2} & \Delta\tilde{\sigma}_{12C,1} \end{pmatrix} \begin{pmatrix} \alpha_1 \\ \alpha_2 \end{pmatrix}$$ (3)

where $Det. = \Delta\tilde{\sigma}_{13C,2}\Delta\tilde{\sigma}_{12C,1} - \Delta\tilde{\sigma}_{12C,2}\Delta\tilde{\sigma}_{13C,1}$ and where the experimental differential absorption coefficient has been corrected by the differential absorption due to $H_2O$:

$$\alpha_i = \alpha_{exp,i} - \alpha_{H2O,i}$$ (4)

where $\alpha_{H2O,i} = C_{H2O}n_{air}(\tilde{\sigma}_{H2O,i} - \tilde{\sigma}_{H2O,0})/(1 + C_{H2O})$.

## 85 2.2 Isotopic ratio $\delta^{13}C$ measurement

The variation of isotopic ratio is measured with respect to a reference, the Vienna Pee Dee Belemnite (VPDB) isotopic ratio 0.011237:

$$\delta^{13}C = \left[\frac{\left(\frac{C_{13}}{C_{12}}\right)_{meas.}}{\left(\frac{C_{13}}{C_{12}}\right)_{VPDB}} - 1\right] \times 1000 ,$$ (5)

Actually, a single $\delta^{13}C$ measurement brings only little information on the carbon cycle for the atmosphere has integrated the
main part of $^{12}CO_2$ and $^{13}CO_2$ variations linked to surface sources and sinks. Rather, usually, many $\delta^{13}C$ measurements are needed in a so-called Keeling plot where $\delta^{13}C$ is reported as a function of the inverse of $^{12}CO_2$ mixing ratio: $\delta^{13}C = f(1/C_{12})$. The extrapolation of $\delta^{13}C$ for high value of $C_{12}$ gives some information about the source or sink of $CO_2$:

$$\delta^{13}C_{source/sink} = \delta^{13}C(C_{12} \rightarrow \infty)$$ (6)

Therefore, the characterization of $CO_2$ source and sink depends not only on $^{12}CO_2$ and $^{13}CO_2$ precision DIAL measurements
but also on the amplitude of $^{12}CO_2$ variations in the atmosphere. In practice, the method needs for plume detection (anthropogenic source mainly) or diurnal cycle acquisition where large variations of $CO_2$ usually appear that are linked to the building of a stratified nocturnal layer and the so-called rectifier effect (Ogée et al. 2003; Widory et al. 2003; Lopez et al. 2013).

## 3 Spectroscopy in the 2 µm band and DIAL wavelengths positioning

$CO_2$ absorption measurements are usually considered in the three spectral regions 1.6, 2 or 4.3 µm. The latter has been preferred in the past given its strong absorption lines (two orders of magnitude larger than at 2 µm) to obtain the highest precision to date (0.02‰ for 400 s of time averaging) on *in situ* $CO_2$ isotope ratio measurement with a spectroscopic technique (Nelson et al. 2008). The 1.6 µm domain has also been considered rather for technical reason (even for $CO_2$ DIAL measurement) although the $CO_2$ low absorption line strength reduced the precision obtained for $\delta^{13}C$ by one order of magnitude (2‰ for 8.7s)
(Kasyutich et al. 2006). Recent in situ cavity-ring-down spectroscopy (CRDS) instrument (PICARRO G2101-i analyzer) uses this spectra domain to obtain a precision better than 0.3‰ (5 min). The 2-µm domain offers a compromise both for the absorption coefficient and the available technical tools (laser, detector) (Andreev et al. 2011). In particular, the 2.05 µm $CO_2$ absorption band has already been used at LMD to make horizontal $CO_2$ profiles in the atmospheric boundary layer with a coherent DIAL (CDIAL) (Gibert et al. 2015). The instrument was able to make a 150 m-15 min $^{12}CO_2$ mixing ratio
measurement with a precision of 0.5% at 500 m. However, usually the $^{13}CO_2$ absorption line intensities are far lower by two orders of magnitude (following the isotopic ratio) than the $^{12}CO_2$ lines. This entails a non-optimal DAOD ($\ll 1$) which reduces significantly the DIAL precision (Bruneau et al. 2006).

Fortunately, in the 2.05 band, the $^{13}CO_2$ line intensities and then ACS are larger by one order of magnitude that mitigates, for one part, the isotope ratio. Figure 1 shows ACS calculated with line intensities from Benner et al. (2016) in the 2.06-µm
spectral region. These were obtained with a multispectrum fitting approach using a modified Voigt line shape to include line mixing and quadratic speed dependence. This database has been improved for air-broadening, air-pressure shift coefficients





and their temperature dependence with recent CRDS lab measurements for $^{12}CO_2$ (Mondelain et al. 2023) and for $^{13}CO_2$ absorption lines (Mondelain et al. 2025) (Table 1). The residuals obtained after the multi-spectrum fit procedure are usually lower than 0.05 % for the absorption lines considered in this study.

Using Table 1 and Figure 1, we are able to calculate typical absorption coefficient due to $^{12}CO_2$ and $^{13}CO_2$ at the chosen wavelengths $\lambda_1$ and $\lambda_2$: ~0.75 km$^{-1}$ and ~0.05 km$^{-1}$ respectively. From Eq. (1), the relative error on DIAL absorption measurement can be roughly written $\sigma(\alpha_i)/\alpha_i \propto 1/(\alpha_i.\delta R.SNR_i)$ where $SNR_i$ is the $\lambda_i$ lidar signal to noise ratio. Therefore, to obtain similar precision on DIAL absorption measurement at $\lambda_1$ and $\lambda_2$, $SNR_2 > 15\ SNR_1$ which represents a huge instrumental effort. Comparing to Gibert et al. (2015) CDIAL set-up used for $^{12}CO_2$ measurement only, range and time

resolution will have then to be reduced and the laser should run at higher pulse repetition frequency to obtain a larger $SNR$.

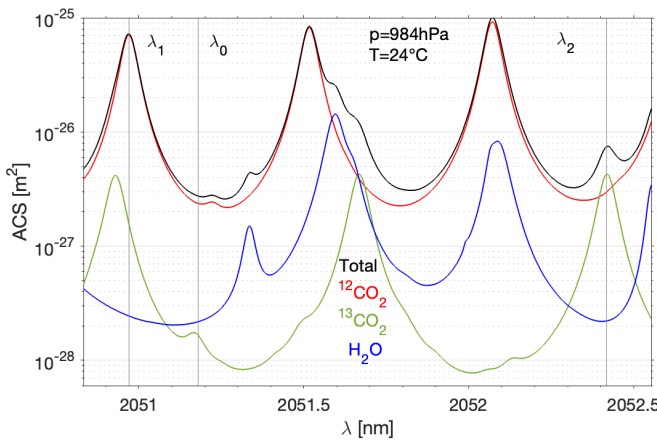

**Figure 1: Absorption cross-section (ACS) for $^{12}CO_2$, $^{13}CO_2$ and $H_2O$ for pressure 984 hPa and temperature 24°C (Voigt profile). $H_2O$ ACS have been multiplied by a factor of 25 with respect to mixing ratios in the atmosphere ($CO_2$: 0.04%, $H_2O$: 1%). $^{13}CO_2$ ACS is multiplied by the VPDB isotopic ratio 0.01118. The DIAL wavelengths chosen in this work are indicated.**


**Table 1: Main $^{12}CO_2$ and $^{13}CO_2$ absorption line parameters used in this study from Mondelain et al. (2023, 2025). One-sigma uncertainties are given in parenthesis in the unit of the last digit. S: line intensity at 296 K weighted by standard isotopic abundance; E": lower energy level of the transition; $\gamma_0$ :air broadening coefficient at 296 K; $n_{\gamma 0}$: temperature dependence exponent; $\delta_0$ :air-pressure shift coefficient at 296 K. Uncertainty for S is 0.003% (one sigma) from Benner et al. (2016).**

| | $\nu$ (cm$^{-1}$) | $\lambda$ (nm) | S (cm/molecule) | E" (cm$^{-1}$) | $\gamma_0$ (m$^{-1}$ atm$^{-1}$) | $n_{\gamma 0}$ | $\delta_0$ (m$^{-1}$ atm$^{-1}$) |
|---|---|---|---|---|---|---|---|
| $^{12}CO_2$ R30 | 4875.7487 | 2050.9670 | 1.5094 x 10$^{-22}$ | 362.7882 | 6.890(2) | 0.744(3) | -0.520(1) |
| $^{12}CO_2$ R24 | 4871.7917 | 2052.6329 | 2.2251 x 10$^{-22}$ | 234.0833 | 7.049(2) | 0.738(3) | -0.493(2) |
| $^{13}CO_2$ P18 | 4872.3027 | 2052.4176 | 9.6179 x 10$^{-24}$ | 133.4456 | 7.360(7) | 0.700(4) | -0.527(2) |
| $^{13}CO_2$ P14 | 4875.8434 | 2050.9272 | 9.5870 x 10$^{-24}$ | 81.9440 | 7.649(6) | 0.69 | -0.510(2) |

## 4 Experimental set-up

The lidar set-up is displayed in Figure 2 and the technical specifications are gathered in Table 2. The laser set-up has been significantly modified since our DIAL $CO_2$ measurements (Gibert et al. 2015, 2018) following recent work on hybrid fiber/bulk amplifier in the 2 µm domain (Lahyani et al. 2020, 2024). Our current emitter uses a seeder module with three narrow linewidth external cavity laser diodes (special model CHEETAH from Sacher Lasertechnik). Both seeder $\lambda_1$ and $\lambda_2$ are locked to $^{12}CO_2$

and $^{13}CO_2$ absorption line centers using two frequency reference systems (FRS) built with low pressured gas cell filled with pure $^{12}CO_2$ at 20 mbar and $^{13}CO_2$ at 5 mbar respectively and a Pound-Drever-Hall (PDH) technique. $\lambda_0$ is locked to $\lambda_1$ using a phase-locking loop and a beat frequency monitoring. The spectral precision and accuracy of each wavelength present an Allan deviation better than 1 MHz at 10 s and 10 min (Gibert et al. 2018). After a 4x1 double stage fibered switch (Agiltron)



(measured cross-talk isolation larger than 45 dB), the seeder power is amplified through a CW Thulium doped fiber amplifier

(TDFA) (model CTFA from Keopsys). The laser pulses are then shaped using an acousto-optic modulator AOM (model Brimrose – 50 MHz). An electro-optical modulator and a double-stage optical fibered-coupled switch (Agiltron – not represented in Fig. 2) is added to reject (by 50 dB) the long settling time CW power after the pulse. The pulse repetition rate is 6 kHz as a compromise to have a large number of samples and keep a sufficient Carrier to Noise Ratio (CNR). The wavelength switch is fixed at 60 Hz (switch every 100 pulses at fixed wavelength) as a compromise to limit switch disturbance

on the measurements and keep identical atmospheric aerosol backscatter signal for the three wavelengths.

A custom Holmium pulsed fibered amplifier (model THALYS-2051nm-0.2W from Cybel) enables to deliver a maximum of 17 W peak power (~ 5 µJ for a 300 ns pulse duration) without parasitic stimulated Brillouin scattering (SBS) effect. The laser pulses are then amplified in a free space multi-pass Ho:YLF amplifier. Six 0.5%-Holmium doped 50-mm long Ho:YLF rods pumped by two 50-W linearly polarized Thulium fiber lasers (model TLR-50-1940-LP from IPG Photonics) are used to obtain

a mean 27 W output power at 2.05 µm (4.5 mJ @ 6 kHz). Both high cross-talk isolation from the double stage optical switch and high side mode suppression from the amplifiers (Lahyani et al. 2024) give an overall spectral purity larger than 45 dB. This laser architecture enables flexible characteristics of the laser: pulse duration, energy and repetition frequency (PRF). A coherent detection with a 50-mm diameter aperture lens, a balanced extended InGaAs photodiode detection (Discovery Semiconductors Inc.) and a four channels data acquisition and real time signal processing system (model Waverider-250 from

LICEL) completes the set-up.

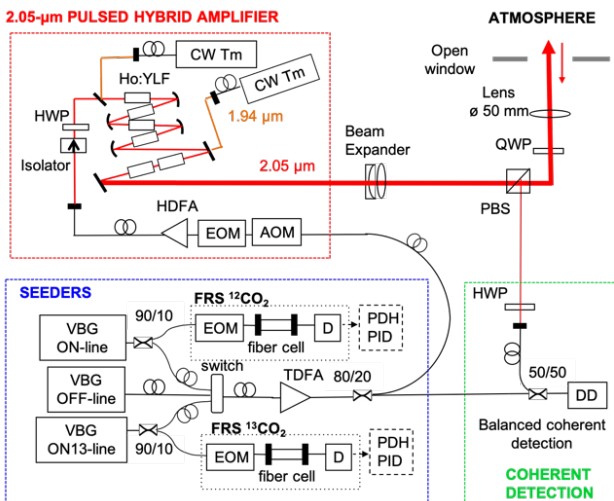

**Figure 2: DIAL set-up for δ$^{13}$C measurement. FRS : frequency reference system, EOM et AOM : electro and acoustic-optical modulators, HDFA and TDFA : Holmium and Thulium fiber amplifiers, VBG :volume Bragg grated laser diode, PDH : Pound-**
**Drever-Hall, HWP and QWP :half and quarter wave plate, PBS : polarizer beam splitter.**

**Table 2: Key specifications of $^{12}$CO$_2$ and $^{13}$CO$_2$ DIAL system. Further details may be found in Gibert et al. 2018, Lahyani et al. 2020 and 2024.**

| Transmitter | | Receiver | |
|---|---|---|---|
| Pulse energy | 4.5 mJ | Lens aperture | 50 mm |
| Pulse duration | 300 ns | LOS elevation | 1.4° |
| PRF | Shot: 6 kHz, Wavelength switch: 60 Hz | Detection Data acquisition system | Extended InGaAs PIN balanced LICEL waverider 4 channels/triggers 250 MHz 14 bit |
| Wavelengths | Ref. ($\lambda_0$) 2051.25 $^{12}$CO$_2$ ($\lambda_1$) 2050.97 $^{13}$CO$_2$ ($\lambda_2$) 2052.42 | Signal Processing | 38.4-m range gate Fourier Transform and 1-s (2000 shots) spectra accumulation Post-processing with Matlab software squarer and Levin estimators |
| Spectral stability | < 1 MHz at 10 min | | |
| Spectral purity | > 45dB | | |





The lidar has been installed on the second floor of LMD building on Ecole Polytechnique campus (10 m height above the
ground). The laser beam is sent quasi-horizontally into the atmosphere through an open window. A CRDS isotope and gas
concentration analyser (PICARRO model G2131-i) made simultaneous continuous measurements of $^{12}CO_2$ and $^{13}CO_2$ mixing
ratio. To complete the dataset, especially to compute the spectroscopic data, meteorological data were collected at the same
height of the LMD lab 200 m away.

## 5 Atmospheric measurements

### 5.1 Absorption coefficient estimates

The signal processing is similar to the one described in Gibert et al. (2015). Real time processing consists in a 38.4-m range
gate Discrete Fourier Transform accumulation over 1 s (2000 shot averaging for each wavelength). Post-processing uses
Matlab software with Squarer and Levin-like estimators to deliver atmospheric backscattered signal power and frequency at
each wavelength (Gibert et al. 2006). Typical lidar profiles averaged over 10 min are displayed in Figure 3.

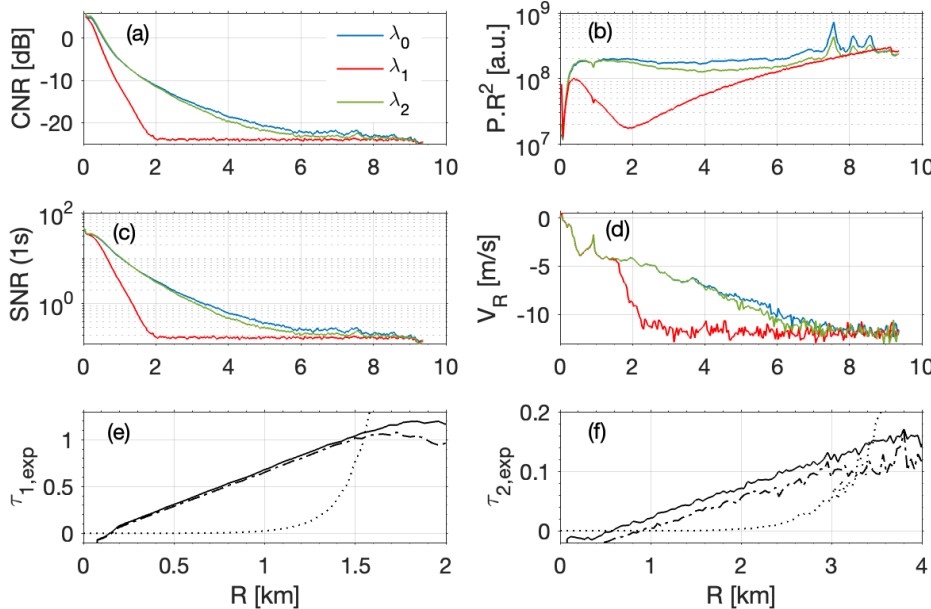

**Figure 3: (a) Carrier to Noise Ratio (CNR) at each wavelength (b) Range-corrected signal (c) Signal to Noise Ratio for 1 s (d)
Radial wind speed (e) DAOD calculated with signals from Levin-like estimator (solid line) and Squarer estimator (dashed and
dotted line) for (e) $\lambda_1$ and (f) $\lambda_2$. The bias on DAOD calculated with Squarer estimates is indicated with the dotted line.**

The CNR, $CNR_i = (P_i - P_{i,B})/P_{i,B}$ where $P_{i,B}$ is the noise power calculated using the integral of the power spectra of the last
range gate of the lidar signal, is estimated for a bandwidth of 30 MHz (Fig. 3a). From the CNR and assuming that $\delta R > c\delta t$,
where $\delta t$ is the pulse duration, the lidar Squarer signal to noise ratio (SNR) may be estimated using (Killinger and Menyuk
1981) (Fig. 3c):

$$SNR_i^{-1} \cong N^{-0.5}(2\delta R/(c\delta t))^{-0.5}(1 + CNR_i^{-1}) \tag{7}$$

where $N$ is the number of averaged shots.

A useful estimate of statistical and systematic error on DAOD (and then absorption) can be written from the SNR (Bosenberg,
1998; Gibert et al. 2008):



$$std(\tau_{i,exp}) = (SNR_i^{-2} + SNR_0^{-2})^{-0.5}/2 \qquad (8)$$

$$\delta(\tau_{i,exp}) = (SNR_0^{-2} - SNR_i^{-2})/4 \qquad (9)$$

Figure 3f shows that the fluctuations on DAOD seems to be far larger for $\tau_{2,exp}$ ($^{13}CO_2$). The reason is that the relative error on $\tau_{2,exp}$, being proportional to the optical depth, is larger by one order of magnitude compared to $\tau_{1,exp}$, despite a larger $SNR_i$. Figures 3e and 3f also show that when the CNR and then the SNR decreases, the calculated DAOD is biased and underestimated as predicted by Eq. (9). Note that Levin-like estimator, which is actually used in this study, is significantly less sensitive to this bias and allows a longer range of measurement (Gibert et al. 2006).

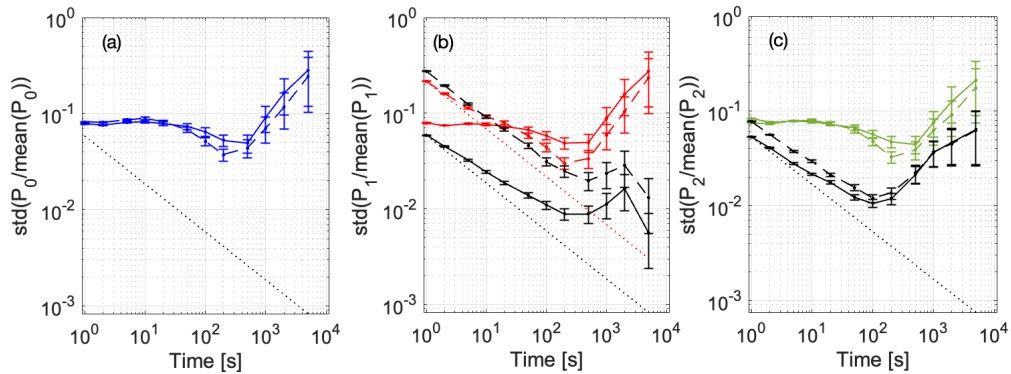


**Figure 4: Allan deviation at R=0.5 km (solid line) and R=1.5 km (dashed line) for (a) $\lambda_0$ (b) $\lambda_1$ (red) and $P_1/P_0$ ratio (black), (c) $\lambda_2$ (green) and $P_2/P_0$ ratio (black). The dotted lines are indicated for the standard white noise deviation ($1/\sqrt{Time}$)**

To optimize the statistical error reduction on the DAOD, we look at the Allan deviation of each signal power at 500 and 1500 m and of the ratios $P_i/P_0$ (Fig. 4). The lidar signal deviation at each wavelength does not decreases as the square root of time

averaging except for very low CNR (CNR< -15 dB), i.e. for $P_1$ deviation at R=1.5 km (Fig. 4b). The averaged lidar signal fluctuations are mainly due to the atmosphere. The Allan deviation of the ratios $P_i/P_0$ shows that these fluctuations are correlated at each wavelength and their deviation decreases almost as the square root of the averaging time (but not entirely suggesting that some uncorrelated noise remains) and caps at few minutes of time averaging. Therefore, in this study, the signal processing consists in averaging the ratios of 1-s lidar signals up to 10 min and then take the logarithm in Eq. (1). The

distance of useful DAOD measurement is chosen to have a negligible bias from Eq. (9). The bias due to potential negative values in the logarithm is also negligible with this process (Tellier et al. 2018). Note that the usual signal processing which consists to average the signals first up to 10 min and then take the ratio and the logarithm entails an increase of the DAOD standard deviation by a factor around two. We suspect that non-random and non-correlated fluctuations between $P_i$ and $P_0$ remains (suspected from detection noise and signal processing) and will need further investigation in the future.

The differential absorption coefficients, $\alpha_{i,exp}$, are obtained with a Matlab bisquare linear fit on the DAOD which limits the weight of outliers with respect to the fitted line (Fig. 5). The regression is made over a distance of 1.3 km for $\alpha_{1,exp}$ ($^{12}CO_2$) and 1.8 km for $\alpha_{2,exp}$ ($^{13}CO_2$) while respecting several conditions: 1) the first points (R< 300 m) that are biased due to parasitic received CW emitted power after the emitted pulse (due to long settling time of the AOM and switch) are discarded, 2) the farthest points that are biased due to low SNR (especially for $\lambda_1$) are also discarded, 3) a reasonable larger number of points

for $\lambda_2$ ($^{13}CO_2$) than for $\lambda_1$ ($^{12}CO_2$) is used to mitigate the large statistical error on $\alpha_{2,exp}$ (due to lower optical depth) while keeping the same air mass (varying number of points have been tested to give credit to this hypothesis).



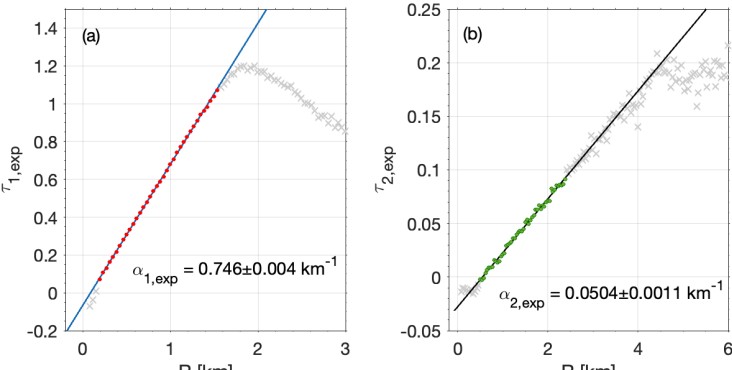

**Figure 5: Differential optical depth for (a) $\lambda_1$ ($^{12}CO_2$) and (b) $\lambda_2$ ($^{13}CO_2$) and typical bisquare linear fit to estimate mean differential absorption coefficient. Colored dotted markers: points used for the fit; cross markers: outliers.**


Given the error on $C_{13}$ absorption coefficient no range-resolved measurement is tested here at higher resolution, i.e. 200 m – 10 min as such processing will only give noisy data where large $CO_2$ source will be hardly identified. Note anyway that range-resolved measurement for $C_{12}$ has been tested and gives similar results as in Gibert et al. (2015). Rather here, our goal is to get a useful precision on $C_{13}$ that will enables us to make a comparison with in situ data. Figure 5 shows an example of mean

differential absorption coefficients estimates. With such conditions, the precisions on $\alpha_{1,exp}$ and $\alpha_{2,exp}$ are around 0.5% and 2%, respectively. Note that an offset spectral locking of $\lambda_1$ with respect to R30 $^{12}CO_2$ absorption line center is possible to have a similar useful distance of measurement than with $\lambda_2$, with the drawback of an increase of the statistical error though. Note also that the optical depths that we used here for $CO_2$ stable isotopologue measurements are similar to the ones used for DIAL $H_2O$ stable isotopologue measurements (i.e. $H_2^{16}O$ and $HD^{16}O$) at 1.98 µm (Hamperl 2022) which enables to make some

comparison in the following sections.

### 5.2 Meteorological data and absorption cross-section calculations

From meteorological data (p, T) and spectroscopic data we calculated the absorption cross section of each molecule ($^{12}CO_2$,

$^{13}CO_2$, $H_2O$) at each wavelength (Fig. 6). The main differential ACS variation is observed at $\lambda_1$ and is due to temperature diurnal cycle (around 0.2% per Kelvin at R30 $^{12}CO_2$ absorption line center) (Gibert et al., 2006).

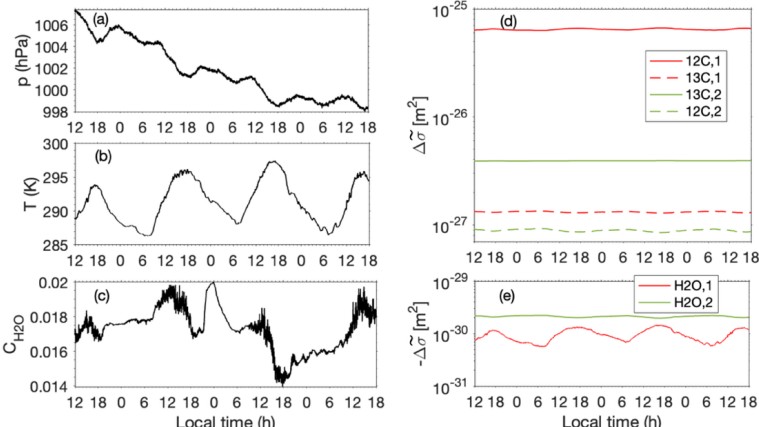

**Figure 6: Meteorological data and differential ACS calculations. (a) Pressure (b) Temperature (c) $H_2O$ mixing ratio (d) Differential ACS at wavelength 1 and 2 for $^{12}CO_2$ and $^{13}CO_2$ absorption lines and (e) for $H_2O$. Note that, as for Fig. 1, $^{13}CO_2$ ACS**
**are multiplied by the VPDB isotopic ratio, i.e. 0.011237. Native in situ meteorological data have a time resolution of 1 min. Differential ACS are computed at the time resolution of Lidar measurements, i.e. 10 min.**



### 5.3 Precision budget


The relative error on mixing ratio estimates results from the different parameters in Eq. (3): 1) the statistical error for absorption coefficient estimates (Fig. 5) and from its correction due to $H_2O$ absorption (Eq. (4)), 2) the uncertainty in differential ACS calculations which comes from uncertainties in meteorological data (mainly temperature) along the Lidar LOS and wavelengths position uncertainty, 3) the uncertainty in dry air density calculations mainly driven by $C_{H2O}$ and T uncertainties

along the Lidar LOS (the error due to pressure fluctuations is neglected). The relative error on each parameter is reported in Table 3. The uncertainties due to temperature fluctuations and water vapor along the line of sight of the Lidar are estimated from in situ data Allan deviation at 10 min, i.e. over the DIAL accumulation time. For the whole acquisition time (Fig. 6), the Allan deviation for T and $C_{H2O}$ is respectively 0.18 K (< 0.07 %) and 1.3 $10^{-4}$ (< 0.9 %). These mean relative errors are used to estimate the errors on differential ACS assuming a Lorentzian absorption line shape and on the dry air density. Note that

the relative error on mixing ratio due to temperature fluctuations in Table 3 accounts both from the differential ACS and the air density (Gibert at al. 2008). The relative error on $\alpha_i$ and then on mixing ratios due to water vapor absorption correction (Eq. 4) is calculated using an upper bound for water vapor absorption contribution in $\alpha_{i,exp}$ from Fig. 6, which amounts to 0.1 % of $\alpha_{1,exp}$ and 1.7 % of $\alpha_{2,exp}$.

**Table 3: Statistical errors on $^{12}CO_2$ ($C_{12}$) and $^{13}CO_2$ ($C_{13}$) mixing ratios.**

| Parameter | Uncertainty (%) | Comment | Reference |
|---|---|---|---|
| $CO_2$ absorption estimates<br>$\quad \alpha_{1,exp}$<br>$\quad \alpha_{2,exp}$ | 0.6<br>3.2 | Resolution: 1.2 km – 10 min<br>Resolution: 1.6km – 10 min | Fig. 5 |
| $H_2O$ absorption correction<br>$\quad \alpha_{H2O,1}$<br>$\quad \alpha_{H2O,2}$ | < 0.0009<br>< 0.015 | From in situ Allan deviation of $C_{H2O}$ at 10 min. Overestimated considering an upper bound of $H_2O$ absorption contribution in $\alpha_{i,exp}$ | Fig. 6 and Eq. (4) |
| Differential ACS:<br>Laser wavelength positioning | < 0.0001 | Overestimated, assuming a wavelength position at absorption line center (R30 ($C_{12}$) and P18 ($C_{13}$) absorption lines) and 1 MHz frequency jitter. | Gibert et al. 2018 |
| Temperature | < 0.01 ($C_{12}$)<br>< 0.08 ($C_{13}$) | From in situ Allan deviation of T at 10 min. Overestimated with calculations at R30 ($C_{12}$) and P18 ($C_{13}$) absorption line centers and including both absorption line intensity and air density fluctuations. | Gibert et al. 2008. Appendix B |
| Dry air density:<br>Water vapor | < 0.02 | From in situ Allan deviation of $C_{H2O}$ at 10 min | Gibert et al. 2008. Appendix B |

Table 3 shows that the uncertainty on mixing ratio estimates is mainly driven by the statistical error for absorption coefficient estimates from lidar backscatter signals. $^{13}CO_2$ mixing ratio estimate is also more sensitive to 1) water absorption correction





due to a smaller absorption optical depth at wavelength 2 and 2) temperature fluctuations due to a less adapted energy level of
the P18 transition than for $^{12}CO_2$ mixing ratio estimate. Although the DIAL system used for water vapor isotopologue
measurements in Hamperl et al. (2022) is very different (direct detection DIAL), similar uncertainties were obtained for
absorption coefficient estimates, i.e. 0.5 % for $H_2O$ and 2.0% for HDO for a time and space resolution of 25 min and 600 m at
a mean distance of 400 m, showing that the measurement performances are driven, in both experiments, by the lowest optical
depth of the stable isotopologues. However, a main difference of the two experiments concerns the spectral characteristics of
the 2 µm emitters as the wavelength position precision contributes significantly in the error budget in Hamperl et al. (2022)
whereas it has a negligible impact in our case.

**5.4 Accuracy budget**

At the current point of our understanding, the systematic errors in DIAL $^{12}CO_2$ and $^{13}CO_2$ mixing ratios comes from: 1) a
statistical bias that is corrected using Eq. (9), 2) approximations in DIAL equation (Eq. (1)) which neglects spectral variation
of aerosol extinction and backscatter coefficients especially for $\lambda_2$ (1.2 nm gap with $\lambda_0$), 3) instrumental characteristics of the
DIAL emitter such as the absolute wavelength positions and the spectral purity, 4) spectroscopic data.

**Table 4: Systematic errors on $^{12}CO_2$ ($C_{12}$) and $^{13}CO_2$ ($C_{13}$) mixing ratios.**

| Parameter | Bias (%) | Comment | Reference |
|---|---|---|---|
| Aerosol scattering change with wavelengths:<br>- Extinction coefficient<br><br>- Backscatter coefficient | $\sim 0.0003\ (\lambda_1)$<br>$\sim 0.016\ (\lambda_2)$<br>$\sim 0.03\ (\lambda_1)$<br>$\sim 0.1\ (\lambda_2)$ | With respect to absorption coefficient estimated at each wavelength. Reference is taken at $\lambda_0$. | Gibert et al. 2008. Appendix A |
| Differential ACS:<br>Laser wavelength positioning | < 0.0001 | Assuming a 1 MHz bias in wavelength position and a Lorentzian absorption line shape | Gibert et al. 2018 |
| Spectral purity | < 0.003 | Mainly driven by to the cross-talk (< -45dB) of the double stage switch | Lahyani et al. 2021, 2024 |
| Spectroscopic data | < 0.03 ($C_{12}$)<br>< 0.09 ($C_{13}$) | Mainly driven by air-broadening coefficient measurements | Mondelain et al. 2023, 2025 |

Given the 1.2 nm wavelength difference in DIAL measurements of $^{13}CO_2$, one may wonder what is the aerosol backscatter
and extinction coefficient differences over such a spectral range. This has actually been studied and quantified in Gibert et al.
(2008), Appendix A. A Mie scattering code for homogeneous spherical particles (Matzler 2002) accounting for typical aerosol
optical depth and size distribution in the suburban area of Palaiseau and a wide range of relative humidity showed an error on
$CO_2$ absorption coefficient of $2\ 10^{-7}\ m^{-1}$ from the backscatter coefficient and $2\ 10^{-9}\ m^{-1}$ from the extinction for a wavelength
difference of 0.3 nm at 2.06 µm. Assuming we can describe the wavelength dependance of extinction (and backscatter
coefficient) with the Angström exponent: $\delta\alpha_i = \alpha_{p,0}\left[1 - (\lambda_0/\lambda_i)^{a^0}\right]$ (Weitkamp 2005). With an assumed suburban $a^0 \sim 1$, we
obtain $\delta\alpha_1 = 1.4\ 10^{-4}\alpha_{p,0}$ for $|\lambda_1 - \lambda_0| = 0.3$ nm and $\delta\alpha_2 = 5.6\ 10^{-4}\alpha_{p,0}$ for $|\lambda_2 - \lambda_0| = 1.2$ nm which keep the numbers
calculated above from Gibert et al. (2008) in the same order of magnitude, i.e. $8\ 10^{-7}\ m^{-1}$ from the backscatter and $8\ 10^{-9}\ m^{-1}$
from the extinction coefficients. These numbers have to be compared to typical $CO_2$ absorption coefficients $\alpha_1$ ($\sim 7.5\ 10^{-4}\ m^{-1}$) and $\alpha_2$ ($\sim5.0\ 10^{-5}\ m^{-1}$) to obtain the relative biases in Table 4. Our bias estimation shows that the aerosol backscatter spectral




difference could have a significant impact on $^{13}CO_2$ mixing ratio estimates (potential bias of 0.1%) if significant aerosol gradients occur in the atmosphere (plume, boundary layer interface). Other systematic errors from the spectroscopic data and DIAL emitter have been quantified as well in Table 4.

In order to quantify the actual biases on $C_{12}$ and $C_{13}$ estimates, we considered the direct problem. In situ differential absorption coefficients for both wavelength 1 and 2 are computed using meteorological, spectroscopic data and in situ PICARRO G2101-i isotopic gas analyzer measurements of $^{12}CO_2$ and $^{13}CO_2$ mixing ratios. Figure 7 shows the comparison between lidar and in situ estimates. A 'good' (with respect to expected representativity error) correlation coefficient of 0.80 and 0.82 are calculated

for $\alpha_1$ and $\alpha_2$, respectively. However, a linear fit shows both a different amplitude of variation and a bias that are not explained up to now, neither by the signal processing nor by the potential biases listed in Table 4. The coefficients of the linear fit remain constant within the uncertainties over the four days of measurements. To pursue the analysis of the lidar measurements, especially on the statistical point of view, we decided to correct the lidar absorption coefficients by these linear fit coefficients keeping in mind that some work will be needed in the future to access to absolute lidar measurements.

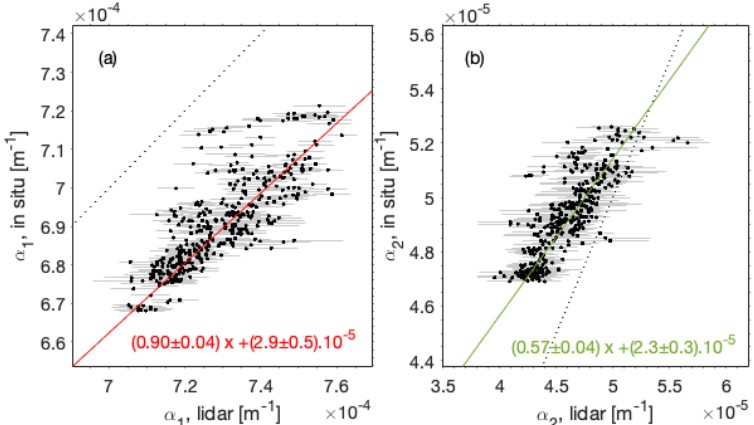


**Figure 7: Comparison between lidar and in situ differential absorption coefficient (from meteorological, spectroscopic and gas analyzer data) for wavelength 1 ($^{12}CO_2$) (a) and 2 ($^{13}CO_2$) (b). The dotted line is for $\alpha_{i,in\ situ} = \alpha_{i,lidar}$.**

## 6 Discussion

### 6.1 Diurnal variations of $^{12}CO_2$ and $^{13}CO_2$ in the atmospheric surface layer


To show the current performances of CDIAL $CO_2$ isotopic measurements, we report almost 70 h of lidar $C_{12}$ and $C_{13}$ mixing ratio measurements in the surface layer at approximately 15 m height above Ecole Polytechnique campus (Fig. 8). The lidar reflectivity and the radial wind speed along the line of sight (LOS) of the lidar are also displayed. The diurnal cycles of lidar $C_{12}$ et $C_{13}$ estimates follow the in situ data with two exceptions on 09/18 (6h) and 09/20 (10h) that correspond to changes in

radial wind speed conditions. Large differences between in situ and lidar measurements are then explained by different spatial representativity. Other major differences are also observed on the evenings, 09/18 (20h) and 09/19 (20h). These correspond likely to anthropogenic plumes (evening road traffic) that are not well mixed during the evening transition and are located hundreds of meters far from the LMD building where is the in situ gas analyzer (R = 0 km). The $^{12}CO_2$ anomaly magnitude of these plumes (10-20 ppm) is similar to what was measured in Gibert et al. (2015). When the atmospheric boundary-layer is

well mixed by turbulence (seen with fluctuated values of $V_R$ along the LOS, 12-18 h in local time) the agreement (within the error bars) between lidar and in situ sensor measurements is excellent as expected.

The natural change in lidar reflectivity and therefore CNR and SNR entails a variation of the maximum distance of measurements (to avoid optical depth biases) and then of the resolution of lidar differential absorption estimates along the 70



h: 1.0-1.3 km for $C_{12}$ and 1.5-1.8 km for $C_{13}$. The precision of lidar mixing ratio estimates changes as well and ranges between 1.3 to 6.4 ppm for $C_{12}$ (median value: 2.3 ppm (0.6%)) and 0.06 to 0.4 ppm for $C_{13}$ (median value: 0.15 ppm (3.2%)).

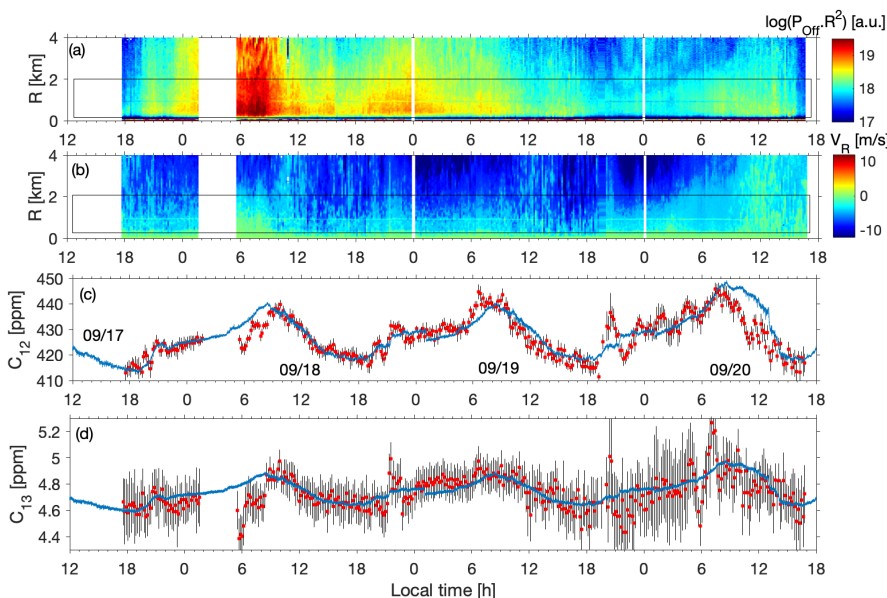

**Figure 8: $C_{12}$ and $C_{13}$ lidar measurements. (a) Lidar reflectivity for wavelength 0 and associated (b) radial wind speed. (c) $C_{12}$ and (d) $C_{13}$ lidar mixing ratio measurements (markers). The solid line corresponds to in situ PICARRO G2101-i analyzer**
**measurements. Space resolution for lidar $C_{12}$ and $C_{13}$ measurements is around 1.0-1.3 and 1.5-1.8 km respectively depending the level of the signal power (panel (a)). Time resolution is 10 min. Error bars are calculated from $\alpha_{i,exp}$ estimates (Fig. 5).**

## 6.2 Current status and future prospective concerning δ¹³C lidar estimates

From Eq. (5), we calculated the $CO_2$ isotopic ratio δ¹³C both for in situ and lidar measurements and report the 70 h data in a Keeling plot (Fig. 9a). A least-squares fit on the in situ data results in a mean $\delta^{13}C_{source} = -26$ ‰ both a result of vegetation respiration and traffic road anthropogenic emissions in this suburban area (Lopez et al. 2003;, Widory et al. 2003). The lidar data gave a similar value although the large uncertainty (260%) prevent us to obtain any conclusion. Using Eq. (5) we can calculate the statistical error on $\delta^{13}C$ :

$$std(\delta^{13}C) \cong \sqrt{\frac{var(C_{12})}{c_{12}^2} + \frac{var(C_{13})}{c_{13}^2}} \qquad (10)$$

From Eq. (10) we understand that the lidar error on $\delta^{13}C$ is driven by the statistical error on $C_{13}$ lidar mixing ratio estimate.

To predict the future performances of optimized lidar system for $\delta^{13}C$ measurement, we made Monte-Carlo numerical simulations assuming the same spread of $1/C_{12}$ in situ data collected during this field experiment and adding some increasing random noise on $\delta^{13}C$ in situ values. The goal was to infer an error threshold of a future optimized lidar system to see

geophysical $\delta^{13}C$ sources (Fig. 9b). Typical $\delta^{13}C_{source}$ anomalies with respect to a mixed air reference of 9 ‰ has been reported on Figure 9b for comparison. The current lidar $\delta^{13}C_{source}$ error is also reported and agrees with Monte-Carlo error simulation. If we keep the same number of points that were used in this study (70 h of measurements with 10 min time resolution) we understand that a threshold lidar error should be around 7‰, 3‰, 2‰ and 1‰ to detect respectively natural gas, coal/fuel/ C-3 plant respiration, marine/geological and C-4 plant respiration $\delta^{13}C_{source}$ anomalies (Widory et al. 2003).

Given that such detection should be made ideally during a single night (10 h acquisition) or for anthropogenic plumes (typical minute time scale), the precision to reach is even more challenging (< 1‰). Concerning our current lidar system, relying on a



coherent detection for $^{13}CO_2$ DIAL measurement, the SNR should be increased by at least 20 dB to obtain a better space resolution and precision around 0.1%, which seems to be technically unreachable with the present coherent detection. However, some hope still exists with direct detection DIAL using internally amplified photodetector such as HgCdTe

avalanche photodiode (Sun et al. 2014; Dumas et al. 2017) or superconducting nanowire single-photon detector (SNSPD) (Yue et al. 2022).

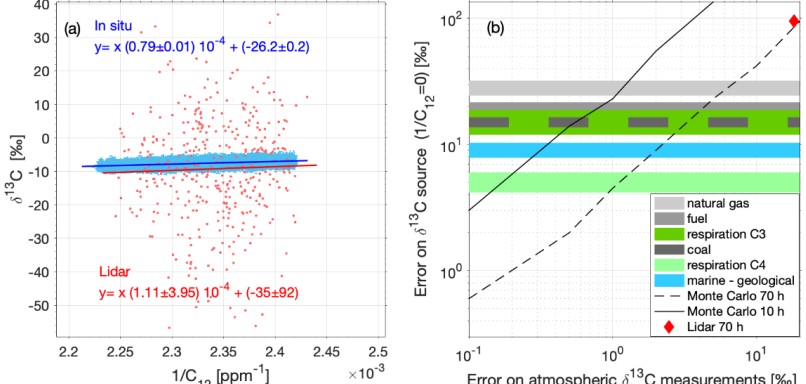

**Figure 9: (a)** Keeling plot for in situ gas analyzer and CDIAL measurements **(b)** Monte Carlo simulations of the $\delta^{13}C_{source}$ error estimate (time resolution is 10 min) using in situ gas analyzer as the truth and random additional noise on $\delta^{13}C$ measurements.

Lidar error on $\delta^{13}C_{source}$ estimate with 70h of measurements is indicated as well as $\delta^{13}C_{source}$ geophysical typical anomalies for natural gas, fuel, coal anthropogenic emissions, C-3 and C-4 plant respiration and marine or geological source with respect to standard atmospheric mixed air $\delta^{13}C$ (9‰).

### 7 Conclusion


A first investigation of range-resolved DIAL for the measurement of $CO_2$ isotopic composition in the atmosphere has been presented. A three wavelengths CDIAL lidar was developed for simultaneous measurements of $^{12}CO_2$ and $^{13}CO_2$ in the 2-µm spectral domain. The spectroscopic database has been updated with recent experimental data with outstanding accuracy and precision. The LMD CDIAL system has also been upgraded since 2015 and relies now on a new 27-W hybrid fiber/bulk

multiple wavelength laser at 2-µm that offers similar performances but a better flexibility with respect to pulse energy, duration and rate tuning. A specific optimized configuration for this study provides a three wavelengths emission with 4.5 mJ, 300 ns and 6 kHz pulses. The CDIAL system was used to make first range-resolved measurements of $^{12}CO_2$ and $^{13}CO_2$ absorption in the atmospheric surface layer above the suburban area in the south of Paris. Typical performances of the instrument (median values along 70h of measurement) with 10 min of time averaging show: (1) a precision around 0.6% for 1.2 km range resolution

for $^{12}CO_2$ mixing ratio (2) a precision around 3.2% for 1.6 km range resolution for $^{13}CO_2$ mixing ratio. In situ co-located gas analyser measurements were used to correct for biases that are explained neither by the spectroscopic database accuracy nor the signal processing and will need further investigation. Then, differences in $^{12}CO_2$ and $^{13}CO_2$ mixing ratio anomalies between in situ and CDIAL made sense as the results of dynamical processes and different sounding representativity in the atmosphere. Once again, the simultaneous radial wind speed ability of the CDIAL system was critical to explain geophysical $CO_2$ variability

in the surface layer linked to surface emissions that are not fully mixed.

Both limited precision and accuracy of the current set-up prevent us to make useful geophysical measurements of the isotopic ratio $\delta^{13}C$ in order to characterize the sources of $CO_2$. Nevertheless, both in situ and CDIAL measurements were used to make a state of the art for current lidar ability to provide $\delta^{13}C$ measurements in the atmosphere with respect to geophysical expected anomalies and to predict the necessary performances of a future optimized instrument. Monte-Carlo simulations

showed that an increase of the instrument SNR by two orders of magnitude is necessary to get a useful geophysical precision better than 1‰ on $\delta^{13}C$. This precision is fully limited by the precision on DIAL $^{13}CO_2$ absorption measurement that suffers



from a small absorption coefficient around 0.05 km$^{-1}$ in the 2-μm spectral domain (compared to 0.75 km$^{-1}$ for $^{12}CO_2$). Despite such increase of SNR seems to be out of range for a coherent DIAL system with reasonable range and time resolution, such performances are still achievable with a direct detection DIAL and this will be tested in a future work.

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
