# Peer review of "$\delta^{13}C$ carbon isotopic composition of $CO_2$ in the atmosphere by Lidar. A preliminary study with a CDIAL system at 2- $\mu$ m."

_EGUsphere, 2025_

## Referee Comment (RC1)

**Review Report for the manuscript, titled "$\delta^{13}C$ carbon isotopic composition of $CO_2$ in the atmosphere by Lidar. A preliminary study with a CDIAL system at 2-µm"**

The manuscript claims the capability of measuring atmospheric carbon dioxide's main isotopologues, $^{12}CO_2$ and $^{13}CO_2$, simultaneously using a coherent differential absorption lidar operating at three wavelengths within the 2-µm spectral region. While the objective of this work is to obtain the range-resolved $CO_2$ isotopic ratio $\delta^{13}C$ (Line 18), the conclusion clearly admits the limitation of obtaining such a ratio due to measurement precision and accuracy, as claimed (Lines 318-319). Therefore, the use of $\delta^{13}C$ in the title is inappropriate.

In fact, careful spectroscopic analysis indicates major fundamental problems in this work, other than precision and accuracy, which implies that obtaining $^{13}CO_2$ measurements is impossible, and thereafter $\delta^{13}C$, with the described setup operating at the stated spectral range. The presented results indicate measuring $^{12}CO_2$ twice using two differential settings, rather than measuring $^{13}CO_2$, which leads to incorrect results interpretation by the authors. Therefore, this manuscript is rejected, and the authors should cautiously review the following issues.

**1- Insufficient Citation:** A significant part of this work was previously published in the 31st International Laser Radar Conference (ILRC), held in Landshut, Germany, 2024, by some of the authors, titled "$\delta^{13}C$ carbon isotopic composition of $CO_2$ in the atmosphere by Lidar". This conference paper is uncited, while it includes the same methodology, spectral analysis and instrumental setup sections presented in this manuscript. In addition, papers presenting similar work by other research teams achieving atmospheric $CO_2$ lidar measurements were uncited, indicating insufficient literature research.

**2- Incorrect Spectral Analysis:** Figure 1 presents the absorption cross-section spectra for $^{12}CO_2$, $^{13}CO_2$ and $H_2O$, with unclear scaling parameters. It is unclear how the curve marked "Total" was obtained. The $^{13}CO_2$ spectral profile is not multiplied by the VPDB isotopic ratio of 0.01118, as stated in the figure caption. Proper spectral analysis is required to justify the lidar measurement presented later. Therefore, the spectral analysis of Figure 1 was reproduced to fully understand the problem.

[Figure]

First, Figure A presents the absorption cross-section spectra for $^{12}CO_2$, $^{13}CO_2$ and $H_2O$ without any scaling factors. The profiles in Figure A were obtained from HITRAN 2020 database using Voigt line model and closely match the profiles presented in Figure 1 except for $H_2O$ magnitude.

*Figure A Absorption Cross Section Calculations using HITRAN 2020.*

Second, Figure B presents the same profiles of Figure A but with amplitude scaling using the same factors claimed in the caption of Figure 1, which include 25 for $H_2O$ and 0.01118 for $^{13}CO_2$. Figure B clearly indicates that the 0.01118 factor was not included in the $^{13}CO_2$ profile of Figure 1 in the submitted manuscript (nor the ILRC paper), whereas the $H_2O$ scaling factor of 25 was included. This leads to the wrong conclusion of the ability to measure $^{13}CO_2$ using on-line (ON13) wavelength shown in Figure 1.

[Figure]

*Figure B Absorption Cross Sections of Figure A scaled according to paper.*

Third, following proper analysis, figure C presents the absorption coefficient spectra for the same molecules, after including the number density and atmospheric abundance of each molecule, as well as the total absorption. From Figure C, it is evident that the total absorption is dominated by $^{12}CO_2$ with some influence from $H_2O$ but without any key contribution from $^{13}CO_2$. The $^{13}CO_2$ absorption coefficient is about 3 orders of magnitude lower than $^{12}CO_2$ and therefore does not contribute to the total absorption. This is due to lower $^{13}CO_2$ absorption strength within this 2-μm region, as presented in Table 1, and lower $^{13}CO_2$ abundance of 0.01118, as presented in the caption of Figure 1.

[Figure]

*Figure C Absorption coefficient spectra using US standard model for water vapor and 727.00 and 4.72 ppm for $^{12}CO_2$ and $^{13}CO_2$, repsctively.*

This indicates that $^{13}CO_2$ is unmeasurable at all using the claimed settings, while $^{12}CO_2$ is measured twice, using two on-line and single off-line wavelengths.

**Improper Interpretation for the Lidar Measurements:** The lidar results presented in Figures 8 (c) and (d) show a high correlation between $C_{12}$ and $C_{13}$ measurements, which is unrealistic. Generally, $^{13}CO_2$ abundance is either uncorrelated or anticorrelated to $^{12}CO_2$, due to the Suess effect. High correlation between $C_{12}$ and $C_{13}$ measurements confirms measuring $^{12}CO_2$ twice using two different spectral settings. $C_{12}$ in Figure 8(c) represents $^{12}CO_2$ measurement with high sensitivity, due to high differential absorption coefficient between ON12 and OFF, of Figure C. $C_{13}$ in Figure 8(d) again represents $^{12}CO_2$ measurement with lower sensitivity, due to lower differential absorption coefficient between ON13 and OFF, of Figure C, but not $^{13}CO_2$ as claimed. That explains the failure to obtain the δ13C ratio by applying Equation 4 to the results of Figure 8 for the same molecule.

**Incorrect Dry Air Terms in Equations:**

In Equations 2 and 3 the term for dry air is given by $(1 - C_{H2O})n_{air}$, which is wrong. The correct term for dry air is $n_{air} / (1 + C_{H2O})$.

---

## Referee Comment (RC2)

Dear Editor,                                                                    September 9, 2025

I have reviewed the manuscript below, which was submitted to the journal *Atmospheric Measurement Techniques:*

"Delta $^{13}$C carbon isotopic composition of $CO_2$ in the atmosphere by Lidar. A preliminary study with a CDIAL system at 2-μm," by authors Fabien Gibert, Dimitri Edouart, Didier Mondelain, Claire Cénac, and Camille Yver.

Overview:

The manuscript describes initial measurements of atmospheric 12CO2 and 13CO2 with a lidar and discusses the results in context of carbon source analysis. The measurements were made from the ground in a nearly horizontal path by using 3 wavelength coherent DIAL lidar that operates in the spectral region between 2050 and 2053 nm. The lidar uses an off-line wavelength, one at the absorption peak of a 12CO2 line, and the other on the absorption peak of a 13CO2 line. The lidar used was a previous coherent DIAL lidar that had been updated with a higher power laser and the new capability to tune to the targeted 13CO2 line. The theory of the CDIAL lidar measurements are reviewed and measurements of 12CO2 and 13CO2 made over a several km long atmospheric path are shown and compared to those from an in-situ sensor. The measurement precisions and accuracies are discussed in the context of those needed for determination of atmospheric CO2 fluxes. The primary measurement challenge is the limited capability to measure 13CO2. This is mainly caused by its weak line absorption due to its small atmospheric concentration of ~ 4 ppm, which is roughly 1% of 12CO2. The manuscript also discusses possible approaches to improve the 13CO2 measurement precision and accuracy.

Findings and Recommendation:

The manuscript addresses work to address an important area to better remotely sense and understand fluxes of carbon between the Earth's surface and atmosphere by measuring the isotopic ratio of atmospheric CO2. It gives a detailed review of the theory of the coherent DIAL lidar measurements. It reports important lidar measurements including those extended in time, comparisons to the in-situ sensors and evaluation of its measurement stability via Allan variance. The manuscript is well written and cites a large number of relevant references.

Although found that some updates needed, I recommend accepting an updated version of this manuscript after the mandatory changes are incorporated.

Mandatory changes:

1. The lidar's wind speed measurements are mentioned a few times, but there is little discussion of them in the manuscript and the wind measurement results aren't shown. If the focus is on CO2 measurements, then I recommend just mentioning the lidar is also capable of wind measurements and give a reference.

2. The lidar's demonstration measurements are made over a few km long nearly horizontal path. They are compared to in situ measurements, which are much more accurate, especially for 13CO2. Given the in-situ sensor's higher accuracy and the lidar's relatively short range, the potential benefits of using a lidar for these type of atmospheric measurements (ie in the atmosphere near the in-situ sensor) is unclear. This needs to be clarified in the introduction and conclusion.

3. The manuscript's introduction needs to be clearer about what accuracies and resolutions are needed for this type of lidar to be useful in determining flux signatures, especially given the small change their fluxes make in Delta CO2.

4. Line 29 the phrase: "devasting consequences for …" please reword using phrasing from a relevant review paper.

5. In Figure 2 the components used to measure the transmitted pulse energies or powers (ie $P_0$) need to be shown. Also it needs to show the key blocks for the signal processing after the detector

6. In Figure 1 the lines for the 3 wavelengths appear faint and need to be darker or clearer

7. For the lidar measurement comparisons to in situ, please give the azimuthal angle of the wind vector (if measured) relative to the lidar's azimuthal pointing angle.

8. In Table 2 the signal processing is described as spectral accumulation, while on Line 177 it references 2000 shot averaging for each wavelength. Please clarify what is meant by spectral accumulation, how it is performed and what is being averaged (the signal or its spectrum)?

9. In Figure 3, (a)-(d) are plotted to 10 km, but the later measurements primarily go to 3 km, while Fig. 3e only to 2 km. Since the most useful range is limited to 3-4 km please replot (a)-(d) accordingly.

10. In Figure 5, the data points are very hard to see. They need to be replotted with symbols that are larger or have more contrast.

11. Both plots in Figure 5 show that both tau values increase linearly with range, Please comment on what that implies for the spatial variability of 12CO2 & 13CO2 in the path. If there is high spatial uniformity of 12CO2 and 13CO2 in the path, then what does that imply about benefits of lidar vs in-situ measurements?

12. In Table 4, it is unclear whether the values are for before or after bias correction. Please clarify.

13. It seems other possible sources of systematic measurement error could be caused by the lidar hardware. Possibilities might include a wavelength dependent response of the optics or detector that measures the Po values for each wavelength or slight changes in the beam pattern from the transmitter for the 3 wavelengths. Please briefly address the possibilities of the lidar hardware being a source of systematic error.

14. In Figure 7 the better known (in situ) measurements are plotted on the y-axis with the lidar values on the x-axis. Since here the primary question here is about the lidar measurements it seems their values should be plotted on the y-axis. Also, if possible, please plot the measurements made on the different days with different colors or symbols.

15. In Figure 9 the dots for the lidar measurements are difficult to see. Please replot in a darker color or larger symbol size.

Recommended changes

1. The manuscript doesn't clearly address the targeted application scenario of this type of lidar. For example, it solely intended for research, or for deployment for single units or a network? Is the targeted use primarily intended for horizontal, slant or vertical path measurements?

2. Line 46, specify what type of gas emissions

3. Line 49 "interesting ways". Do the authors mean simpler or more affordable?

4. Line 53 "outstanding information." Do the authors mean new information?

5. Line 62 "confronted" Do the authors mean compared to?

6. Line 78, 1st two equations, the equal signs did not get type set correctly

7. Line 113. "the ACS are larger by one order of magnitude" Do the authors mean compared to those in the 1.6 um band?

8. On page 5 please comment on why the wavelengths are switched at 60 Hz.

9. On page 5 the diameter of the lens used for the common transmit receive path is given as 50 mm. Diffraction limited lens are commercially available with much larger diameters and it seems that using one would improve the lidar's CNRs and the measurement precisions. Please comment on why this lens diameter was chosen.

---

## Author Comment (AC1)

**Response to the Review Report # 2 for the manuscript, titled "δ13C carbon isotopic composition of CO2 in the atmosphere by Lidar. A preliminary study with a CDIAL system at 2-µm"**

The manuscript claims the capability of measuring atmospheric carbon dioxide's main isotopologues, 12CO2 and 13CO2, simultaneously using a coherent differential absorption lidar operating at three wavelengths within the 2-µm spectral region. While the objective of this work is to obtain the range-resolved CO2 isotopic ratio δ13C (Line 18), the conclusion clearly admits the limitation of obtaining such a ratio due to measurement precision and accuracy, as claimed (Lines 318-319). Therefore, the use of δ13C in the title is inappropriate.

From our measurements, d13C has been indeed calculated in this manuscript as claimed in the title despite a large uncertainty. Section 6 of the manuscript gives also some guidelines to measure geophysical features of d13C in the atmosphere with a lidar. We would like to keep d13C in the title as this is the topic of our work.

In fact, careful spectroscopic analysis indicates major fundamental problems in this work, other than precision and accuracy, which implies that obtaining 13CO2 measurements is impossible, and thereafter δ13C, with the described setup operating at the stated spectral range. The presented results indicate measuring 12CO2 twice using two differential settings, rather than measuring 13CO2, which leads to incorrect results interpretation by the authors. Therefore, this manuscript is rejected, and the authors should cautiously review the following issues.

**The analysis and the conclusion of the reviewer is based on a misunderstanding of HITRAN2020 database.** In HITRAN2020, the standard isotopic ratio 0.01118 is already included in isotopologue line intensity (https://hitran.org/docs/definitions-and-units/). Then Figure A of the reviewer proves that our spectral domain is fully relevant to measure 12CO2 and 13CO2 as claimed in the manuscript.
Please re-consider your review and conclusion with these facts.

**1- Insufficient Citation:** A significant part of this work was previously published in the 31st International Laser Radar Conference (ILRC), held in Landshut, Germany, 2024, by some of the authors, titled "δ13C carbon isotopic composition of CO2 in the atmosphere by Lidar". This conference paper is uncited, while it includes the same methodology, spectral analysis and instrumental setup sections presented in this manuscript. In addition, papers presenting similar work by other research teams achieving atmospheric CO2 lidar measurements were uncited, indicating insufficient literature research.

Corrected. Citations related to CO2 absorption measurements by lidar have been added in the introduction as suggested by the reviewer :

"Several lidar teams have been interested in measuring $CO_2$ absorption with DIAL systems since almost twenty years with precursor work using DIAL systems in the 2 µm spectral band and coherent detection (Koch et al. 2004, Gibert et al. 2006) and more recent works (Gibert et al. 2015). The spectral band of 1.6 µm has also been considered with coherent DIAL (Yu et al. 2024) or direct detection DIAL, using the advantage of low noise internally amplified photodetector (Shibata et al. 2017; Yue et al. 2022; Stroud et al. 2023) although the obtained precision was limited by the ten times lower $CO_2$ absorption optical depth at such wavelength. "

**2- Incorrect Spectral Analysis:** Figure 1 presents the absorption cross-section spectra for 12CO2, 13CO2 and H2O, with unclear scaling parameters. It is unclear how the curve marked "Total" was obtained. The 13CO2 spectral profile is not multiplied by the VPDB isotopic ratio of 0.01118, as stated in the figure caption. Proper spectral analysis is required to justify the lidar measurement presented later. Therefore, the spectral analysis of Figure 1 was reproduced to fully understand the problem.

We agree that Figure 1 and associated caption is unclear and could lead to a somewhat misunderstanding. To clarify the picture, we remove the total ACS curve and we modify the caption :
« Absorption cross-section (ACS) for 12CO2, 13CO2 and H2O for pressure 984 hPa and temperature 24°C (Voigt profile). Isotopic ratio is taken into account. The DIAL wavelengths chosen in this work are indicated. »
New Figure 1 is below:

[Figure]

First, Figure A presents the absorption cross-section spectra for 12CO2,13CO2 and H2O without any scaling factors. The profiles in Figure A were obtained from HITRAN 2020 database using Voigt line model and closely match the profiles presented in Figure 1 except for H2O magnitude.

Figure A corresponds to former Figure 1 except for H2O where ACS has been multiplied by a factor 25 assuming mean H2O mixing ratio is 1% and CO2 mixing ratio is 0.04%. To clarify Figure 1, standard ACS from the database (without factor 25 due to relative CO2/H2O abundance in the atmosphere) has been displayed.

Second, Figure B presents the same profiles of Figure A but with amplitude scaling using the same factors claimed in the caption of Figure 1, which include 25 for H2O and 0.01118 for 13CO2. Figure B clearly indicates that the 0.01118 factor was not included in the 13CO2 profile of Figure 1 in the submitted manuscript (nor the ILRC paper), whereas the H2O scaling factor of 25 was included.This leads to the wrong conclusion of the ability to measure 13CO2 using on-line (ON13) wavelength shown in Figure 1.

Figure B is a nonsense. Isotopic ratio is already included in HITRAN ACS (https://hitran.org/docs/definitions-and-units/).

Third, following proper analysis, figure C presents the absorption coefficient spectra for the same molecules, after including the number density and atmospheric abundance of each molecule, as well as the total absorption. From Figure C, it is evident that the total absorption is dominated by $12CO_2$ with some influence from $H_2O$ but without any key contribution from $13CO_2$. The $13CO_2$ absorption coefficient is about 3 orders of magnitude lower than $12CO_2$ and therefore does not contribute to the total absorption. This is due to lower $13CO_2$ absorption strength within this 2-µm region, as presented in Table 1, and lower $13CO_2$ abundance of 0.01118, as presented in the caption of Figure 1.
This indicates that $13CO_2$ is unmeasurable at all using the claimed settings, while $12CO_2$ is measured twice, using two on-line and single off-line wavelengths.

Figure C is also a nonsense. When isotope ratio is included in the line intensity, the absorption of the isotopologue (i.e. here $13CO_2$) is calculated with the abundance of the main isotopologue (i.e. here $12CO_2$ ~400 ppm) and not with the abundance of $13CO_2$ ~4 ppm.
The analysis/conclusion of the reviewer is obviously wrong when you know that isotopic ratio is included in HITRAN ACS.

**Improper Interpretation for the Lidar Measurements:**
The lidar results presented in Figures 8(c) and (d) show a high correlation between C12 and C13 measurements, which is unrealistic.

Both in situ and lidar measurements are displayed in Figure 8. The reviewer comment concerns $12CO_2$ and $13CO_2$ measurements provided by the in situ PICARRO gas analyzer which is a reference sensor for such measurements. I don't understand what « unrealistic » means here.

Generally, $13CO_2$ abundance is either uncorrelated or anticorrelated to $12CO_2$, due to the Suess effect. High correlation between C12 and C13 measurements confirms measuring $12CO_2$ twice using two different spectral settings.

No, this is definitively wrong as shown by the in situ PICARRO measurements in Figure 8 and Figure 9a. Despite a strong correlation in PICARRO C12 and C13 measurements, there is still some information in d13C represented in the Keeling plot. These d13C measurement have to be extrapolated to the origin (1/CO2$\rightarrow$ 0) to get some information of sources and sinks. Please see Widory, D. and Javoy, M.: The carbon isotope composition of atmospheric CO2 in Paris, Earth and Planetary Science Letters, 215, 289-298, doi: 10.1016/S0012-821X(03)00397-2, 2003

C12 in Figure 8(c) represents $12CO_2$ measurement with high sensitivity, due to high differential absorption coefficient between ON12 and OFF, of Figure C. C13 in Figure 8(d) again represents $12CO_2$ measurement with lower sensitivity, due to lower differential absorption coefficient between ON13 and OFF, of Figure C, but not $13CO_2$ as claimed. That explains the failure to obtain the $\delta13C$ ratio by applying Equation 4 to the results of Figure 8 for the same molecule.

Wrong remark due to wrong understanding of HITRAN database.

**Incorrect Dry Air Terms in Equations:**
In Equations 2 and 3 the term for dry air is given by $(1 - CH_2O)n_{air}$, which is wrong. The correct term for dry air is $n_{air} / (1 + CH_2O)$.

Correct. It was a typo that has been corrected in the preprint version of the paper. In the calculations we used for the dry air: nair / (1 + CH2O). $C_{H2O}$ and $C_{12}$ being dry air mixing ratio. Equations 2 and 3 have already been corrected in the preprint version.

---

## Author Comment (AC2)

**Response to reviewer #3**

First, I would like to thank Reviewer #3 for his constructive review of my manuscript.
I answered to the different points below and modification in the new manuscript are indicated with line number.

I have reviewed the manuscript below, which was submitted to the journal *Atmospheric Measurement Techniques:*

"Delta 13C carbon isotopic composition of CO2 in the atmosphere by Lidar. A preliminary studywith a CDIAL system at 2-µm," by authors Fabien Gibert, Dimitri Edouart, Didier Mondelain, Claire Cénac, and Camille Yver.

Overview:
The manuscript describes initial measurements of atmospheric 12CO2 and 13CO2 with a lidar and discusses the results in context of carbon source analysis. The measurements were made from the ground in a nearly horizontal path by using 3 wavelengths coherent DIAL lidar that operates in the spectral region between 2050 and 2053 nm. The lidar uses an off-line wavelength, one at the absorption peak of a 12CO2 line, and the other on the absorption peak of a 13CO2 line. The lidar used was a previous coherent DIAL lidar that had been updated with a higher power laser and the new capability to tune to the targeted 13CO2 line. The theory of the CDIAL lidar measurements are reviewed and measurements of 12CO2 and 13CO2 made over a several km long atmospheric path are shown and compared to those from an in-situ sensor. The measurement precisions and accuracies are discussed in the context of those needed for determination of atmospheric CO2 fluxes. The primary measurement challenge is the limited capability to measure 13CO2. This is mainly caused by its weak line absorption due to its small atmospheric concentration of ~ 4 ppm, which is roughly 1% of 12CO2. The manuscript also discusses possible approaches to improve the 13CO2 measurement precision and accuracy.
Findings and Recommendation:
The manuscript addresses work to address an important area to better remotely sense and understand fluxes of carbon between the Earth's surface and atmosphere by measuring the isotopic ratio of atmospheric CO2. It gives a detailed review of the theory of the coherent DIAL lidar measurements. It reports important lidar measurements including those extended in time, comparisons to the in-situ sensors and evaluation of its measurement stability via Allan variance. The manuscript is well written and cites a large number of relevant references. Although found that some updates needed, I recommend accepting an updated version of this manuscript after the mandatory changes are incorporated.

Mandatory changes:

1. The lidar's wind speed measurements are mentioned a few times, but there is little discussion of them in the manuscript and the wind measurement results aren't shown. If the focus is on CO2 measurements, then I recommend just mentioning the lidar is also capable of wind measurements and give a reference.
Yes. We added the following sentence:
L183. Doppler frequency shift is used to infer the radial wind speed at each wavelength (Gibert et al. 2015).

2. The lidar's demonstration measurements are made over a few km long nearly horizontal path. They are compared to in situ measurements, which are much more accurate, especially for 13CO2. Given the in-situ sensor's higher accuracy and the lidar's relatively short range, the potential benefits of using a lidar for these types of atmospheric measurements (ie in the

atmosphere near the in-situ sensor) is unclear. This needs to be clarified in the introduction and conclusion.

Note that in the introduction of the reviewed version of the manuscript we added some information about the interest to make d13C measurement by a range-resolved DIAL system. Note that the lidar short range ~km could be significantly improve (10 km) with an off-center 12CO2 R30 absorption line locking of lidar wavelength 1.

L52. "the horizontal profiling and 2-D mapping of $\delta$13C field above the surface by Lidar will bring outstanding information on sources/sinks pattern and origin."
L 53. "Even the vertical profiling will help to characterize the local/ long distance transport of CO2 in a similar way as for stable water vapor isotopologue Lidar measurements (Hamperl et al. 2022)."

We also added the interest for a future CO2 space lidar mission:

L55. "Ultimately, the capability of 13CO2 lidar measurements opens the way to a global monitoring of 13CO2 from space using the IPDA technique to improve global carbon inversion systems (Chen et al. 2017)."

3. The manuscript's introduction needs to be clearer about what accuracies and resolutions are needed for this type of lidar to be useful in determining flux signatures, especially given the small change their fluxes make in Delta CO2.

Searched precision and resolution for $\delta$13C DIAL have been added in the introduction from Widory and Javoy, 2003.

L47 "However, hundreds of meters from the source, these anomalies are reduced to sub 1 ‰ variations due to efficient mixing of the atmosphere (Widory and Javoy, 2003). "

L56. "However, 1 ‰ precision with hundreds of meters range resolution has not yet been reached for CO2 DIAL system. "

L62 "more recent works (Gibert et al. 2015) reaching a precision of 0.5% with 150-m and 15-min range and time resolution, respectively, close to what is needed for $\delta$13C observations."

4. Line 29 the phrase: "devasting consequences for …" please reword using phrasing from a relevant review paper.
The sentence has been modified:
"CO2 is the main anthropogenic greenhouse gas responsible for the current global warming. In 2024, its global annual average concentration in the atmosphere has reached more than 420 ppm and the global mean near surface temperature is 1.5 °C above the 1850-1900 average, with significant consequences for present and future life on planet Earth (WMO 2025, IPCC 2023)."

5. In Figure 2 the components used to measure the transmitted pulse energies or powers (ie P0) need to be shown. Also it needs to show the key blocks for the signal processing after the detector
P0 is not the transmitted power but the backcattered power at reference non-absorbed wavelength (indicated L82). For a DIAL measurement, an accurate measurement of the transmitted power is not necessary.

Signal processing scheme has been included in Figure 2 and Figure 2 caption has been modified.

6. In Figure 1 the lines for the 3 wavelengths appear faint and need to be darker or clearer
Corrected. Lines are black now.

7. For the lidar measurement comparisons to in situ, please give the azimuthal angle of the wind vector (if measured) relative to the lidar's azimuthal pointing angle.
Figure 8 has been modified. Horizontal wind direction at 10 m has been added and is compared to lidar line-of-sight azimuth.

8. In Table 2 the signal processing is described as spectral accumulation, while on Line 177 it references 2000 shot averaging for each wavelength. Please clarify what is meant by spectral accumulation, how it is performed and what is being averaged (the signal or its spectrum)?
Range-gate DFTs at a given distance R are averaged over a number N of shots. In coherent detection, only spectra are used and accumulated.
Table 2 has been clarified: "Real time 38.4-m range gate Discrete Fourier Transform (DFT) spectrum accumulated over 1-s (2000 shots) "

9. In Figure 3, (a)-(d) are plotted to 10 km, but the later measurements primarily go to 3 km, while Fig. 3e only to 2 km. Since the most useful range is limited to 3-4 km please replot (a)-(d) accordingly.
Figure 3 has been modified.

10. In Figure 5, the data points are very hard to see. They need to be replotted with symbols that are larger or have more contrast.
Figure 5 data points have been magnified and contrast has been improved.

11. Both plots in Figure 5 show that both tau values increase linearly with range, Please comment on what that implies for the spatial variability of 12CO2 & 13CO2 in the path. If there is high spatial uniformity of 12CO2 and 13CO2 in the path, then what does that imply about benefits of lidar vs in-situ measurements?
Although Figure 5 shows that tau values increase linearly with range, this does just mean that 12CO2 and 13CO2 concentrations are almost constant over the lidar LOS. But this actually does not mean anything because CO2 concentration fluctuations are in the order of the ppm range which is lower than 1%. 1% variation of the slope (concentration is proportional to the slope and not tau in DIAL measurement) cannot be seen in Figure 5. Therefore, Figure 5 cannot be used to discuss the benefits of lidar vs in situ measurements. For those who are interested in, Gibert et al. (2015) range-resolved measurements answer to this question.

12. In Table 4, it is unclear whether the values are for before or after bias correction. Please clarify.
Table 4 gives just an assessment of potential systematic errors on 12CO2 (C12) and 13CO2 (C13) mixing ratios. The only bias that is corrected is the one due to Eq. (9) (indicated at L295). Table 4 caption has been clarified.

13. It seems other possible sources of systematic measurement error could be caused by the lidar hardware. Possibilities might include a wavelength dependent response of the optics or detector that measures the Po values for each wavelength or slight changes in the beam pattern from the transmitter for the 3 wavelengths. Please briefly address the possibilities of

the lidar hardware being a source of systematic error.

Concerning P0, please see the answer of point 5. There was some misunderstanding here. Concerning the source of systematic error this has been added L321:

"These biases are likely to be due to the long settling time of the emitted pulse shaped by the AOM (Fig. 2). As seen in Figure 3a, a part of the emitted pulse (and the remaining power after it) is reflected by the optics after the polarizer (Fig. 2) which may create a different bias on atmospheric backscattered power at each wavelength."

14. In Figure 7 the better known (in situ) measurements are plotted on the y-axis with the lidar values on the x-axis. Since here the primary question here is about the lidar measurements it seems their values should be plotted on the y-axis. Also, if possible, please plot the measurements made on the different days with different colors or symbols.

Figure 7 has been modified with lidar measurements in the y-axis and in situ in the x-axis. Bi-square regression has been made again.

I tried to put the measurements of different days with different markers and colors but I gave up because there was a significant overlap between points that prevents the reader to distinguish efficiently the different points of the different days. Then, I decided to keep the same markers and colors for the whole dataset.

15. In Figure 9 the dots for the lidar measurements are difficult to see. Please replot in a darker color or larger symbol size.

Figure 9 has been modified and the size of the markers has been magnified.

Recommended changes

1. The manuscript doesn't clearly address the targeted application scenario of this type of lidar. For example, it solely intended for research, or for deployment for single units or a network? Is the targeted use primarily intended for horizontal, slant or vertical path measurements?

The different targets of such instrumental development have been clarified (see mandatory point 1.) with additional references.

2. Line 46, specify what type of gas emissions

Natural gas emission has been specified L47

3. Line 49 "interesting ways". Do the authors mean simpler or more affordable?

I meant "simpler". Corrected L51

4. Line 53 "outstanding information." Do the authors mean new information?

Yes "new information". Corrected L55

5. Line 62 "confronted" Do the authors mean compared to?

Yes "compared to", Corrected L74

6. Line 78, 1st two equations, the equal signs did not get type set correctly

Corrected. See new equation (3) L92. These equations have been significantly modified after reviewer 1 comments.

7. Line 113. "the ACS are larger by one order of magnitude" Do the authors mean compared to those in the 1.6 um band?

No. The $^{13}CO_2$ ACS are larger by order of magnitude than expected by the isotope ratio (two orders of magnitude). The sentence has been modified by: "ACS are larger by one order of

magnitude than expected by the isotope ratio, that mitigates, for one part, their non-optimal DAOD. " L125

8. On page 5 please comment on why the wavelengths are switched at 60 Hz.
As written L160, "The wavelength switch is fixed at 60 Hz (switch every 100 pulses at fixed wavelength) as a compromise to limit switch disturbance on the measurements (switch cross-talk is limited to 30 dB) and keep identical atmospheric aerosol backscatter signal for the three wavelengths."
Experiments have been achieved where the optical switch was used in different configurations and we noted an impact (biais on the slope) on DIAL measurement when the switch was used at high frequency (On-Off wavelength switch at 2 kHz). To do that we operate the system in (Off-Off) configuration to see if we could measure a 0 optical depth. This issue disappears when we operated the switch at lower frequency, i.e. 60 Hz.
At such frequency, we checked that the atmospheric backscatter remained the same.

9. On page 5 the diameter of the lens used for the common transmit receive path is given as 50 mm. Diffraction limited lens are commercially available with much larger diameters and it seems that using one would improve the lidar's CNRs and the measurement precisions. Please comment on why this lens diameter was chosen.
Numerical simulations were made to calculate heterodyne efficiency with different lens diameter 50 – 100 mm. 100 mm diameter lens only improves the heterodyne efficiency for a distance longer than 2 km. In addition, coherent detection with larger aperture is more sensitive to the turbulent conditions close to the surface ($C_n^2$) (large $C_n^2$ entails a decrease of coherent radius and then increases the number of speckles for a given aperture which decreases the heterodyne efficiency). In our conditions, i.e. horizontal measurement over a distance of 2 km, 50 mm diameter lens offers the best results. This was confirmed by experimental tests.
We added " (optimal heterodyne efficiency for horizontal measurement close to the surface over a distance of 2 km)" L170